# Prognostic Biomarker SPOCD1 and Its Correlation with Immune Infiltrates in Colorectal Cancer

**DOI:** 10.3390/biom13020209

**Published:** 2023-01-20

**Authors:** Lin Gan, Changjiang Yang, Long Zhao, Shan Wang, Zhidong Gao, Yingjiang Ye

**Affiliations:** 1Department of Gastroenterological Surgery, Peking University People’s Hospital, Beijing 100044, China; 2Beijing Key Laboratory of Colorectal Cancer Diagnosis and Treatment Research, Laboratory of Surgical Oncology, Peking University People’s Hospital, Beijing 100044, China

**Keywords:** SPOCD1, colorectal cancer, prognosis, immune microenvironment

## Abstract

The biological role of the spen paralogue and orthologue C-terminal domain containing 1 (SPOCD1) has been investigated in human malignancies, but its function in colorectal cancer (CRC) is unclear. This study investigated the association between SPOCD1 expression and clinicopathological features of CRC cases, as well as its prognostic value and biological function based on large-scale databases and clinical samples. The results showed that the expression level of SPOCD1 was elevated in CRC, which was generally associated with shortened survival time and poor clinical indexes, including advanced T, N, and pathologic stages. Multivariate Cox regression analysis showed that elevated SPOCD1 expression was an independent factor for poor prognosis in CRC patients. Functional enrichment analysis of SPOCD1 and its co-expressed genes revealed that SPOCD1 could act as an oncogene by regulating gene expression in essential functions and pathways of tumorigenesis, such as extracellular matrix organization, chemokine signaling pathways, and calcium signaling pathways. In addition, immune cell infiltration results showed that SPOCD1 expression was associated with various immune cells, especially macrophages. Furthermore, our findings suggested a possible function for SPOCD1 in the polarization of macrophages from M1 to M2 in CRC. In conclusion, SPOCD1 is a promising diagnostic and prognostic marker for CRC, opening new avenues for research and treatment.

## 1. Introduction

Colorectal cancer (CRC) poses a significant threat to human health as a prevalent kind of malignant tumor affecting the digestive system. Globally, the death rate from CRC is the second highest among all cancerous tumors, and the incidence is the third highest [1]. Presently, the molecular processes underlying the progression of CRC are not completely clear. CRC therapy often involves a multimodal approach, with surgery playing a central role. The prognosis for CRC has improved over time because of advancements in surgical technique and the introduction of new medications, but the effectiveness is still subpar. It has been proved that early diagnosis and treatment of tumors are crucial in improving CRC prognosis [2]. The discovery of new markers for the early detection and improved prognosis of CRC is, thus, an important requirement.

The spen paralogue and orthologue C-terminal domain containing 1 (SPOCD1), also known as PPP1R146, encodes a protein belonging to the transcription factor S-II family (TFIIS), which is indispensable for the regulation of RNA synthesis. Recent research has shown that SPOCD1 performed a vital function in executing piRNA-directed de novo DNA methylation [3]. To date, the potential functions of SPOCD1 in carcinogenesis have attracted a great deal of attention. Remarkably, some research has investigated SPOCD1-related biological roles in human malignancies. Zhu et al. [4] detected SPOCD1 variants linked to greater gastric cancer risk. They found that SPOCD1 was subjected to expression in gastric cancer and that silencing this gene severely hampered the formation of xenograft tumors in nude mice and the ability of gastric cancer cells to proliferate and invade. Liu et al. [5] found that in ovarian cancer, SPOCD1 may have a function in tumorigenesis by stimulating the PI3K/AKT pathway to inhibit cell apoptosis. SPOCD1 was also shown to upregulate PTX3, thus enhancing glioma cells’ ability to proliferate and metastasize [6]. Moreover, SPOCD1 serves as a pro-oncogenic factor in osteosarcoma by enhancing cell proliferation and suppressing cell apoptosis [7]. Van der Heijden et al. [8] constructed a five-gene expression signature including SPOCD1, which had a 79% sensitivity and an 86% specificity (AUC = 0.83) for distinguishing between individuals with the progressive condition regarding T1G3 bladder cancer and those without. These studies uncovered SPOCD1’s fundamental significance in malignancies and may point the way toward a more nuanced knowledge of the protein’s certain functions and molecular mechanisms. Suppression of SPOCD1 has shown promise as a potential cancer therapy method, and it is widely regarded as a promising treatment and prognostic target for cancer. Nonetheless, the potential role of SPOCD1 in CRC is undocumented and poorly characterized.

Consequently, we used the Gene Expression Omnibus (GEO) and The Cancer Genome Atlas (TCGA) databases to compare SPOCD1 expression in CRC and normal samples. To additionally elucidate the clinical features of SPOCD1 in CRC patients, we next assessed the relationships between SPOCD1 expression profiles and patient-specific clinical features and survival outcomes. We also evaluated how SPOCD1 mRNA levels correlate with the tumor immune microenvironment (TIME). These findings offer a novel insight into the involvement of SPOCD1 in CRC advancement and the anti-tumor immune response.

## 2. Materials and Methods

### 2.1. Data Acquisition and Preprocessing

To gather the RNA-Seq data of 647 CRC patients along with 51 normal tissues on gene expression and relevant clinical data, we retrieved the TCGA database for colon adenocarcinoma (COAD) and rectum adenocarcinoma (READ) in Xiantao academic online analysis tool (https://www.xiantao.love/ (accessed on 5 August 2022)). GSE10950, GSE110224, and GSE83889 were selected in this study and acquired from GEO (http://www.ncbi.nlm.nih.gov/geo (accessed on 5 August 2022)).

### 2.2. Survival Analysis

We performed survival analyses utilizing the Kaplan-Meier (KM) technique and the log-rank test, with the median SPOCD1 expression level serving as the cut-off value. The influence of clinicopathologic variables on patient outcomes was examined by means of univariate and multivariate Cox regression analyses. Prognostic variables with *p*-value < 0.1 in univariate analysis were incorporated into subsequent multivariate analysis. Subsequently, multivariate Cox analysis was employed to detect independent prognostic markers.

### 2.3. Enrichment Analysis

ClusterProfiler tool in R tool (3.6.3) was adopted to conduct a gene set enrichment analysis (GSEA), along with Gene Ontology (GO) and Kyoto Encyclopedia of Genes and Genomes (KEGG) analyses. Analysis using GO involved cellular components (CC), molecular functions (MF), and biological processes (BP). GSEA is a computer tool for assessing the statistical power and concordance of variations in two biological states based on an a priori-determined collection of genes [9]. We employed the adjusted *p*-value and normalized enrichment score (NES) to further categorize the enriched pathways in each phenotype. When deciding on a reference gene set for the hallmark pathway, the h.all.v7.2.symbols.gmt [Hallmarks] was chosen. For the GO terms, the reference gene set was chosen as C5. All.v7.2.symbols.gmt [Gene ontology] was selected as the reference gene set of the GO term. Significant enrichment criteria of gene sets were determined at adjusted *p* < 0.05 and false discovery rate (FDR) < 0.25.

### 2.4. Analyses Immune Infiltration

The ESTIMATE (Estimation of Stromal and Immune cells in Malignant Tumours using Expression data) algorithm was employed to calculate the stromal and immune scores of CRC [10]. Using the GSVA program in R [11] we undertook a single-sample gene GSEA (ssGSEA) to analyze the literature-based association between SPOCD1 and the hallmark genes of 24 different kinds of immune cells [12]. The Tumor Immune Estimation Resource (TIMER) database was also applied to examine the link between SPOCD1 and infiltrating immune cells, especially CD8+ T cells, neutrophils, CD4+ T cells, macrophages, B cells, and dendritic cells [13,14]. Spearman’s correlation analysis was used to probe the link between SPOCD1 expression and these immune cells, and the Wilcoxon rank-sum test was adopted to make a comparison of the infiltration status of immune cells between the low- and high-SPOCD1 expression groups.

### 2.5. Patients and Clinical Specimens

The ethical committees of the Peking University People’s Hospital granted their approval for our research. This research was completed in conformity with the guidelines stipulated in the Declaration of Helsinki, and all patients submitted their written informed consent. Eligible patients comprised those with CRC who received surgical treatment at the Peking University People’s Hospital from July 2022 to September 2022. In total, 20 patients were recruited in the study, including one case of stage I, 10 cases of stage II, four cases of stage III, and five cases of stage IV. Before undergoing surgical treatment, none received radiotherapy or chemotherapy. The diagnosis of adenocarcinoma was done for all patients by the use of pathological assessment. To perform immunohistochemistry (IHC) analysis, surgically excised samples of tumor tissues and adjoining normal tissues that were more than 5 cm distant from the tumor were fixed in formalin at a concentration of 4% before embedding them in paraffin blocks.

### 2.6. Immunohistochemistry (IHC) Staining

Sections of colorectal cancer and adjacent normal tissue that had been fixed in paraffin were dewaxed in dimethylbenzene and then rehydrated in ethanol at gradient concentrations. The sodium citrate solution was used in antigen retrieval by microwave heating at 95 °C. Thereafter, after applying H_2_O_2_ at a concentration of 3% for 10 m, the endogenous peroxidase was successfully inhibited. After being covered with a blocking solution, the tissue slices were then treated for an hour with 10% fetal bovine serum. After adding the primary antibody SPOCD1 (1:400; 22243-1-AP), the slices were incubated at 4 °C for the whole night. After that, the slices of tissue were placed in a dark chamber and treated with an HRP-labeled universal anti-rabbit secondary antibody. After counterstaining with hematoxylin, the tissue slices were dried and mounted after the immunostaining was detected with diaminobenzidine (DAB). The level of SPOCD1 expression was evaluated by two different pathologists, using the proportion of positively stained cells and the intensity of the staining. The proportion of positive cells was classified as indicated: 0 (0–10%), 1 (10–40%), 2 (40–70%), and 3 (>70%), while the intensity of the staining was scored as 1 (weak), 2 (moderate), and 3 (strong). The immunohistochemistry (IHC) scores of the SPOCD1 expression profiles were computed by adding the proportion of positive cells to the intensity of the staining. The ultimate definition of the SPOCD1 expression is as specified: low expression, 0–3 points, and high expression, 4–6 points.

### 2.7. Immunofluorescence Staining

We selected 10 CRC tissue samples for immunofluorescent staining. The sections were dewaxed, hydrated, and inactivated endogenous peroxidase activity first. Then after antigen repair, the processed sections were incubated with primary antibodies for SPOCD1 (1:400; 22243-1-AP) and CD163 (1:100; ab156769) at 4 °C for 48 h. Then sections were washed in TBS and incubated with appropriate secondary antibodies overnight at 4 °C. Finally, the percentage of positive cells was counted, and the results were confirmed by two pathologists.

### 2.8. Prediction of Drug Sensitivity

The 60 different kinds of cancer cells that are included in the Center for Cancer Research of the National Cancer Institute (NCI) serve as the primary basis for the CellMiner database [15,16]. Firstly, we visited the homepage of the CellMiner database, then selected the Download Data Sets, got access to the data download portal, and selected medication data (Compound activity: DTP NCI-60) and RNA expression data (RNA: RNA-seq). The NCI-60 cell line is the gold standard line for providing a large representative sample of cancer cells for use in evaluating new cancer treatments. After retrieving data on SPOCD1 gene expression, the R function was used to determine the correlation coefficient between SPOCD1 and medications.

### 2.9. Statistical Analysis

R 3.6.3 was employed for the analysis of the statistical data derived from TCGA. Wilcoxon rank-sum test and Wilcoxon signed-rank test were applied in comparing SPOCD1 expression patterns between the tumors and normal samples. Welch’s one-way ANOVA with Bonferroni’s post hoc test (or the *t*-test) was applied to ascertain if there was a remarkable association between SPOCD1 expression and the clinical-pathological variables. Pearson’s chi-square test was employed to elucidate how various clinicopathological variables influenced SPOCD1 expression. If there are categories in the chi-square test that do not meet the conditions of theoretical frequency > 5 or total sample size > 40, choose Fisher’s exact probability method. To determine SPOCD1’s predictive ability, a Kaplan-Meier (KM) curve was constructed. Moreover, Progression Free Interval (PFI), Disease Specific Survival (DSS), and Overall Survival (OS) were analyzed via univariate and multivariate Cox regression analyses to ascertain the prognostic significance of SPOCD1 expression along with other clinical-pathological parameters. Each significant parameter from the univariate analysis was subjected to the subsequent multivariate analysis. The pROC software was applied to execute the receiver operating characteristic (ROC) analysis of SPOCD1. The area under the curve (AUC) values between 0.5 and 1.0 were recorded, suggesting a discriminating capability of 50% to 100%. A two-tailed probability value of ≤0.05 was significant for all tests.

## 3. Results

### 3.1. The mRNA and Protein Expression Levels of SPOCD1 Are Upregulated in CRC

Using information from the TCGA database, we examined the SPOCD1 gene expression profiles between various human malignant cancers and normal samples. SPOCD1 mRNA expression level was substantially elevated in malignant cancers as opposed to their matched normal counterparts, particularly in the case of stomach adenocarcinoma (STAD), breast cancer (BRCA), head and neck squamous cell carcinoma (HNSC), colon adenocarcinoma (COAD), and rectum adenocarcinoma (READ) (Figure 1A,B). In particular, SPOCD1 was subjected to upregulation in CRC (Figure 1C,D). Three more GEO datasets (GSE10950, GSE110224, and GSE83889) were used to validate SPOCD1 expression. In the three different datasets, SPOCD1 was upregulated in CRC tissues in contrast with adjacent normal samples (Figure 1F–H). The AUC of the ROC curve for SPOCD1 expression was 0.906 (95% CI = 0.874–0.938), indicating the excellent predictive potential for distinguishing tumor tissues from normal tissues (Figure 1E). Afterward, we employed an IHC assay to assess SPOCD1 expression in 20 tumors and adjacent normal tissue obtained from the CRC samples at Peking University People’s Hospital. As shown by the data, SPOCD1 IHC scores were considerably elevated in tumors as opposed to normal tissues (Figure 1I,J).

### 3.2. SPOCD1 Expression Correlates with the Tumor Pathological Stage of CRC

We subsequently correlated SPOCD1 expression patterns with clinical and pathological features in a sample set consisting of 644 CRC patients from TCGA. The expression of SPOCD1 was shown to be substantially linked to the T stage, but not N, M, pathologic stage as per Welch one-way ANOVA followed by Bonferroni’s post-hoc test (Figure 2A–D). At the same time, the median SPOCD1 expression value was used as the cut-off valve for categorizing the samples into low- and high-expression groups. Following this, we evaluated the link between SPOCD1 expression in cancerous tissue and the clinical and pathological variables as follows: clinical stage, age at diagnosis, distant metastasis, N stage, T stage, and gender. Table 1 provides an overview of the associations between clinicopathological factors and SPOCD1 expression in 644 CRC patients. When compared to individuals with lowered SPOCD1 levels, those with elevated SPOCD1 levels were at a more advanced pathological stage (*p* = 0.037), N stage (*p* = 0.043), and T stage (*p* = 0.014). In addition, IHC staining also revealed increased expression of SPOCD1 in stage III and IV CRC compared with stage I and II CRC (Figure 2I,J). No significant association was identified between the level of SPOCD1 expression and any other clinicopathological characteristics.

### 3.3. High SPOCD1 Expression Predicts Poor Prognosis in CRC Patients

We probed the prognostic relevance of SPOCD1 in anticipating the PFI, OS, and DSS outcomes of all CRC cases. Patients with lowered SPOCD1 levels fared better in terms of OS, DSS, and PFI, as determined by KM survival analysis (Figure 2E–G). Univariate and multivariate analyses were undertaken to ascertain if SPOCD1 expression and any other clinicopathological features may independently function as risk markers for patients with CRC. From the univariate analysis, unfavorable OS was significantly linked to age > 65 years old (*p* < 0.001), T3 and T4 stage (*p* = 0.004), N1 and N2 stage (*p* < 0.001), distant metastasis (*p* < 0.001), TNM III/IV stage (*p* < 0.001), and high SPOCD1 expression level (*p* = 0.011). Independent predictors of OS as determined by multivariate analysis included age (*p* < 0.001), M stage (*p* < 0.001), TNM III/IV stage (*p* = 0.002), and SPOCD1 levels (*p* = 0.038) (Table 2, Figure 2H). Moreover, univariate analysis confirmed that a shortened DSS duration was significantly linked to T3 and T4 stage (*p* = 0.002), N1 and N2 stage (*p* < 0.001), distant metastasis (*p* < 0.001), and high SPOCD1 expression level (*p* = 0.007). Moreover, multivariate analysis illustrated that the M stage (*p* < 0.001) and TNM stage (*p* = 0.015) were independent predictors of DSS (Table 3). Also, univariate analysis illustrated that T3 and T4 stage (*p* < 0.001), N1 and N2 stage (*p* < 0.001), distant metastasis (*p* < 0.001) TNM III/IV stage (*p* < 0.001), and high SPOCD1 expression level (*p* = 0.001) were significantly linked to decreased PFI. In addition, the M stage (*p* < 0.001) independently functioned as a predictor of PFI as determined by multivariate analysis (Table 4).

### 3.4. SPOCD1 May Be Involved in Malignant Progression of CRC

To investigate the biological roles of SPOCD1 in CRC, we first identified the top 50 positively significant genes associated with SPOCD1, as shown in the heatmap (Figure 3A), in which solute carrier family 11 member 1 (SCL11A1) was the gene most correlated with SPOCD1 expression (Figure 3B). Interestingly, similar to SPOCD1, patients with low SLC11A1 expression had better survival outcomes in terms of OS than those with high SLC11A1 expression (Figure 3C). We then performed GO and KEGG analysis for these genes. GO term annotation revealed that co-expressed genes with a correlation coefficient > 0.6 of SPOCD1 were enriched mainly in extracellular structure and matrix organization, endoplasmic reticulum lumen, collagen fibril organization, collagen trimer, collagen-containing extracellular matrix, etc. (Figure 3D). As for KEGG pathway analysis, the genes related to SPOCD1 expression were mainly enriched in ECM-receptor interaction, phagosome, protein digestion and absorption, etc. (Figure 3D). Furthermore, GSEA from the KEGG database showed that high SPOCD1 expression was also associated with apical junction, calcium signaling pathway, and chemokine signaling pathway (Figure 3E–G). Together the results of these gene enrichment analyses indicate that SPOCD1 likely participates in the malignant progression of CRC.

### 3.5. SPOCD1 Is Associated with CRC Immune Infiltration

For CRC, the ESTIMATE, stromal, and immune scores were all determined utilizing the ESTIMATE method. These scores showed a substantial positive link to SPOCD1 expression (Figure 4A–C). SPOCD1 expression was correlated with ssGSEA-measured infiltration levels of different kinds of immune cells as per Spearman correlation, including DC cells (r = 0.417, *p* < 0.001), activated DC (r = 0.302, *p* < 0.001), plasmacytoid DC (r = 0.312, *p* < 0.001), immature DC (r = 0.481, *p* < 0.001), B cells (r = 0.136, *p* < 0.001), eosinophils (r = 0.289, *p* < 0.001), macrophages (r = 0.636, *p* < 0.001), mast cells (*p* = 0.420, *p* < 0.001), neutrophils (r = 0.459, *p* < 0.001), NK cells (r = 0.500, *p* < 0.001), NK CD56 bright cells (r = 0.083, *p* = 0.035), NK CD56 dim cells (*p* = 0.187, *p* < 0.001), cytotoxic cells (r = 0.330, *p* < 0.001), T cells (r = 0.216, *p* < 0.001), CD8 T cells (r = 0.187, *p* < 0.001), T effector memory (r = 0.387, *p* < 0.001), T follicular helper (r = 0.300, *p* < 0.001), T gamma delta (r = 0.270, *p* < 0.001), Th1 cells (r = 0.455, *p* < 0.001), Treg (r = 0.296, *p* < 0.001) and Th17 cells (r = −0.233, *p* < 0.001) in CRC (Table 5, Figure 4D). Notably, macrophages, which are key players in regulating angiogenesis, cancer cell proliferation, metastasis, immunosuppression, extracellular matrix remodeling, checkpoint blockade immunotherapy as well as chemoresistance, were found to have the strongest link to SPOCD1 expression (Figure 4E) [17]. Moreover, SPOCD1 expression was shown to correlate with macrophages and other immune cells (dendritic cells, CD8+ T cells, neutrophils, and CD4+ T cells) by searching the TIMER database, suggesting that these cell types may participate in the immunological immersion process of colorectal cancer (Figure 4F–K). We additionally probed the link between the expression of SPOCD1 and the markers of diverse subtypes of macrophages. The results disclosed that the correlation between SPOCD1 and M2 macrophages (CD163, r = 0.569, *p* < 0.001, CD206: r = 0.507, *p* < 0.001) was far greater than M1 macrophages (CD68, r = 0.296, *p* < 0.001) (Figure 5A–C). Immunofluorescence staining also showed that CD163 was highly expressed in CRC tissues with high expression of SPOCD1 (*p* < 0.05) (Figure 5D,E).

Furthermore, we contrasted SPOCD1 expression to those of other genes involved in immunological regulation, such as those involved in immune stimulation, immune suppression, and the major histocompatibility complex (MHC) molecule. SPOCD1 has a strong relationship with the vast majority of the immunological microenvironment’s targets (Figure 6A–C). The transforming growth factor-beta 1 (TGFB1) has the most significant correlation with SPOCD1 (Figure 6D). This may be linked to the mechanism through which upregulated SPOCD1 contributes to a dismal prognosis of CRC. To elucidate the association of SPOCD1 expression with immune cell migration, we analyzed chemokines and chemokine receptors (Figure 7A,B). The research found a link between the expression of SPOCD1 and chemokines and receptors, with the most relevant chemokine and receptor being CCL18 (r = 0.553, *p* < 0.001) and CCR1 (r = 0.538, *p* < 0.001), respectively (Figure 7C). An elevated SPOCD1 level could enhance immune cell migration since it correlates with chemokines and chemokine receptors. These data showed SPOCD1’s involvement in CRC immune infiltration.

### 3.6. SPOCD1 Expression Is a Potential Indicator of Drug Sensitivity

We obtained drug sensitivity and gene expression data from CellMiner, eliminated FDA-unapproved medicines, and determined the association between SPOCD1 expression and drug sensitivity in R utilizing the cor. test and correlation function. In this study, we focused on the 12 leading medications linked to SPOCD1. Several pharmaceuticals were shown to have an association with SPOCD1 expression, including Bleomycin, Cabozantinib, Rapamycin, Cediranib, Everolimus, BLU-667, Abiraterone, Temsirolimus, Zoledronate, Quizartinib, and Caffeic acid (Figure 8). Tumor cells were more sensitive to these medicines when the SPOCD1 level was upregulated. These findings point to SPOCD1 expression as a potential indicator of drug sensitivity.

## 4. Discussion

CRC, a prevailing cancer of the digestive tract, is a leading contributor to cancer-related deaths globally [1]. Although there have been advancements in CRC treatment, this is still one of the deadliest cancers with a low five-year chance of survival [18]. Consistent CRC-focused clinical and fundamental science research has helped advance diagnosis and therapy and has had tangible therapeutic effects. However, the exact mechanisms of progression and invasion of CRC remain to be elucidated. Inadequate molecular markers for early diagnosis and prognostic evaluation are still a problem. Researching novel CRC prediction and prognosis biomarkers have become a pressing clinical need. In particular, the discovery of a powerful diagnostic biomarker is extremely important. High-throughput sequencing and gene microarrays have recently uncovered a mechanism for mining public datasets for new biomarkers. In this research, we visited public databases for information on molecules that influence CRC prognosis and analyzed potential processes implicated in CRC development.

Recently, mounting data suggested that SPOCD1 may be a driving force for carcinogenesis. Many studies demonstrated the upregulation of SPOCD1 in bladder cancer [8], gastric cancer [4], glioma [6], and osteosarcoma [7]. As far as we are aware, there are few reports on the involvement of SPOCD1 in carcinogenesis in CRC. We began our investigation by assessing SPOCD1 mRNA expression in a TCGA pan-cancer sample. The mRNA levels of SPOCD1 were discovered to be substantially elevated in CRC and several other malignancies as opposed to normal tissues. Three more GEO datasets and IHC analysis of a surgical sample validated the overexpression of SPOCD1 in CRC. In addition, a reduced chance of survival was observed in patients with elevated SPOCD1 expression levels. The CRC patients’ pathological stage, N stage, and T stage were substantially linked to SPOCD1 upregulation. Additionally, the PFI, DSS, and OS lengths of CRC patients with elevated SPOCD1 levels were all considerably lower. The expression of the SPOCD1 level was one of the independent predictors for OS. These findings highlighted the possible function of SPOCD1 in CRC tumor growth, and we validated its application as a prognostic and diagnostic marker for the disease.

The GO/KEGG enrichment and GSEA analyses also suggest that SPOCD1’s putative mechanism is related to the formation of the extracellular matrix, the organization of collagen fibrils, and the calcium signaling pathway. Notably, GSEA findings demonstrated the involvement of the chemokine signaling pathway in the SPOCD1 expression phenotype, demonstrating a putative role for SPOCD1 in the control of chemokines within the immune microenvironment. Interestingly, SLC11A1, the molecule with the strongest correlation with SPOCD1 expression, was previously implicated in immune responses [19]. These findings suggested that SPOCD1’s contribution to CRC’s dismal prognosis may be attributed to its ability to regulate the immune microenvironment.

Numerous research reports have illustrated that the CRC-related immune microenvironment performs a pivotal function in the disease’s onset and progression [20]. Research on immunotherapy methods, particularly the immune checkpoint inhibitor (ICI), for CRC has shown surprising clinical success in recent years, contributing to the revolution in cancer immunotherapy paradigms [21]. This therapy is effective for only a proportion of individuals. The next step is to identify a novel biomarker linked to immune infiltrates and molecular pathways which could underlie the immunotherapy’s success.

Our research showed that SPOCD1 expression is linked to several different immunological marker genes and immune cells. Previous research has shown that the immune milieu performs a fundamental function in tumorigenesis [22]. The intricate tumor microenvironment (TME) is necessary for tumor growth and metastasis throughout the whole tumorigenic process. To investigate the potential involvement of SPOCD1 in the immune response, we monitored TIICs associated with SPOCD1 expression using the ssGSEA and TIMER databases. As per the findings of the research, SPOCD1 expression had a direct connection to macrophages. Macrophages, which are a part of the mononuclear phagocytic system, have an essential contribution to make in the regulation of innate immunity, as well as tissue inflammation and homeostasis [23]. Classically activated M1 (proinflammatory) macrophages and alternatively activated M2 (anti-inflammatory) macrophages are the two polarization states that macrophages may attain via specialized differentiation [24]. It has been shown that M1 macrophages may strengthen the immune system’s capacity in fighting against tumors. Contrastingly, M2 macrophages are pro-tumor because they either directly or indirectly stimulate angiogenesis and immunosuppression [25]. We discovered that SPOCD1 was more strongly linked to M2 macrophages than M1 macrophages. These results suggested that SPOCD1 might promote CRC by inducing M2 macrophages.

To explore the mechanism by which SPOCD1 regulates the immune microenvironment, we studied the relationship between SPOCD1 and immunostimulatory genes, immunoinhibitory genes, and MHC molecules. The greatest association was discovered between SPOCD1 and the immunoinhibitory gene TGFB1. The TGF-β protein, which is encoded by the TGFB1 gene, is at the core of the TGF-β pathway, which has been linked to the onset of several malignancies [26]. Extensive research has shown that the TGFB1 gene contributes to the onset and advancement of CRC. TGFB1 is responsible for the promotion of the epithelial–mesenchymal transition as well as metastases in CRC [27]. Upregulation of TGFB1 has also been shown to dampen the body’s natural defenses against tumor cells, which in turn enhances tumorigenesis [28]. Interestingly, TGFB1 was shown to stimulate macrophage polarization toward the M2 subtype [29].

We also studied the association between SPOCD1 expression and chemokines and chemokine receptors. CCL18 is primarily generated and released into the TME by tumor-associated macrophages (TAMs), where it promotes invasion and metastasis [30]. CCR1 is involved in recruiting macrophages to tumor sites [31]. This further confirmed the previous results.

Finally, we scrutinized the associations between the sensitivity of diverse drugs and SPOCD1. The findings highlighted the positive link between SPOCD1 and the sensitivity of anti-tumor drugs such as Bleomycin [32], Cabozantinib [33], Cediranib [34], etc., highlighting that CRC patients with elevated SPOCD1 levels may respond to these medications and suggesting SPOCD1 may be utilized as a predictive marker of the curative efficacy, which is useful for attaining appropriate treatment of CRC.

Certainly, additional investigation is warranted to probe the biological functions of SPOCD1 and the fundamental modulatory mechanisms in CRC onset and advancement. Specifically, more fundamental and clinical trials are necessary to fully validate the biological involvement of SPOCD1 in CRC.

## 5. Conclusions

Our results show that SPOCD1 is expressed at a high level in CRC tissues and that this expression is linked to a highly advanced illness stage and a worse prognosis for CRC patients. Thus, SPOCD1 could serve as a viable marker of CRC patients’ prognostic status.

## Figures and Tables

**Figure 1 biomolecules-13-00209-f001:**
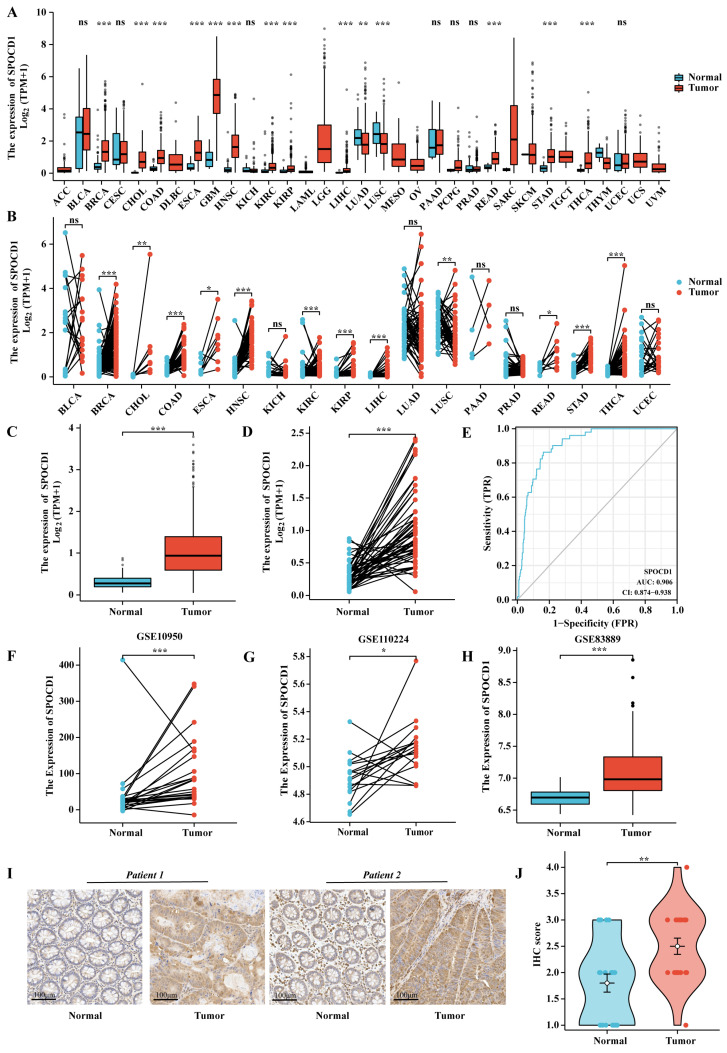
mRNA and protein expression levels of SPOCD1 in pan-cancer and colorectal cancer (CRC) versus normal samples. (**A**,**B**) SPOCD1 mRNA expression was upregulated in pan-cancer tissues compared with normal tissues based on TCGA. (**C**,**D**) SPOCD1 mRNA expression was upregulated in CRC compared with normal tissues based on TCGA. (**E**) ROC analysis of SPOCD1 in the diagnosis of CRC. (**F**–**H**) SPOCD1 mRNA expression was upregulated in CRC compared with normal tissues based on GEO. (**I**,**J**) IHC staining showing SPOCD1 protein expression was upregulated in CRC samples compared with normal tissues. (ns represents no significance, * *p* < 0.05, ** *p* < 0.01, *** *p* < 0.001).

**Figure 2 biomolecules-13-00209-f002:**
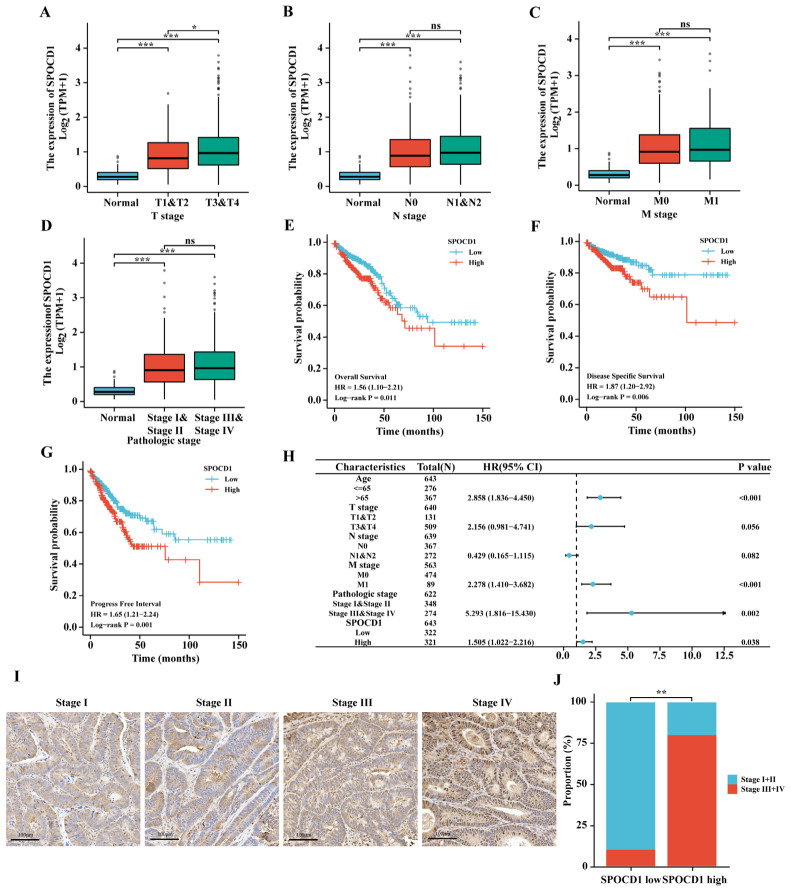
Associations between SPOCD1 expression and clinicopathologic characteristics and its prognostic significance in CRC based on TCGA database. SPOCD1 expression was significantly associated with T stage (**A**) but not with N stage (**B**), M stage (**C**), and pathological stage (**D**). (**E**–**G**) The Kaplan-Meier curves show that CRC patients with a higher expression of SPOCD1 had a shorter overall survival time, disease-specific survival time, and progress-free interval time. (**H**) Multivariate Cox analyses of factors affecting the overall survival of CRC patients in the TCGA database. (**I**) immunohistochemistry (IHC) staining showing SPOCD1 protein expression in CRC samples of the different pathological stages. (**J**) Stacked histogram shows the proportion of CRC at different stages in the high and low SPOCD1 expression groups. (ns represents no significance, * *p* < 0.05, ** *p* < 0.01, *** *p* < 0.001).

**Figure 3 biomolecules-13-00209-f003:**
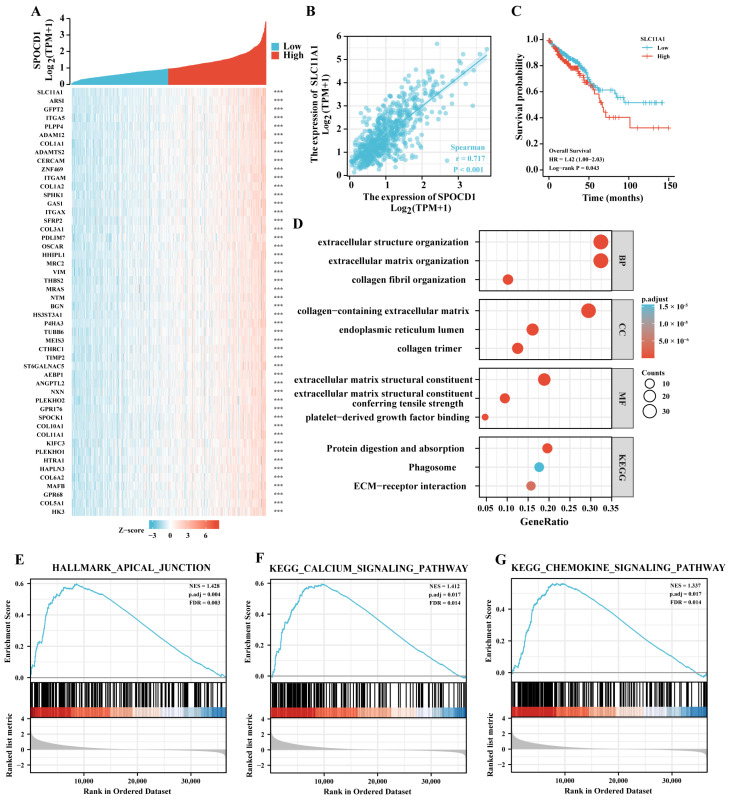
Co-expression gene analysis and subsequent GO and KEGG enrichment analysis of SPOCD1. (**A**) Heatmap illustrating the top 50 genes positively correlated with SPOCD1 in CRC. Red represents high expression, and blue represents low expression. (**B**) Correlation analysis of SPOCD1 and SCL11A1. (**C**) The Kaplan-Meier plots for overall survival for CRC patients according to the difference SLC11A1 expressions. (**D**) GO and KEGG pathway analysis of SPOCD1 co-expressed gene with a correlation coefficient > 0.6 in CRC. (**E**–**G**) GSEA enrichment plots showed positive correlations of SPOCD1 co-expression genes with the apical junction, calcium, and chemokine signaling pathway (*** *p* < 0.001).

**Figure 4 biomolecules-13-00209-f004:**
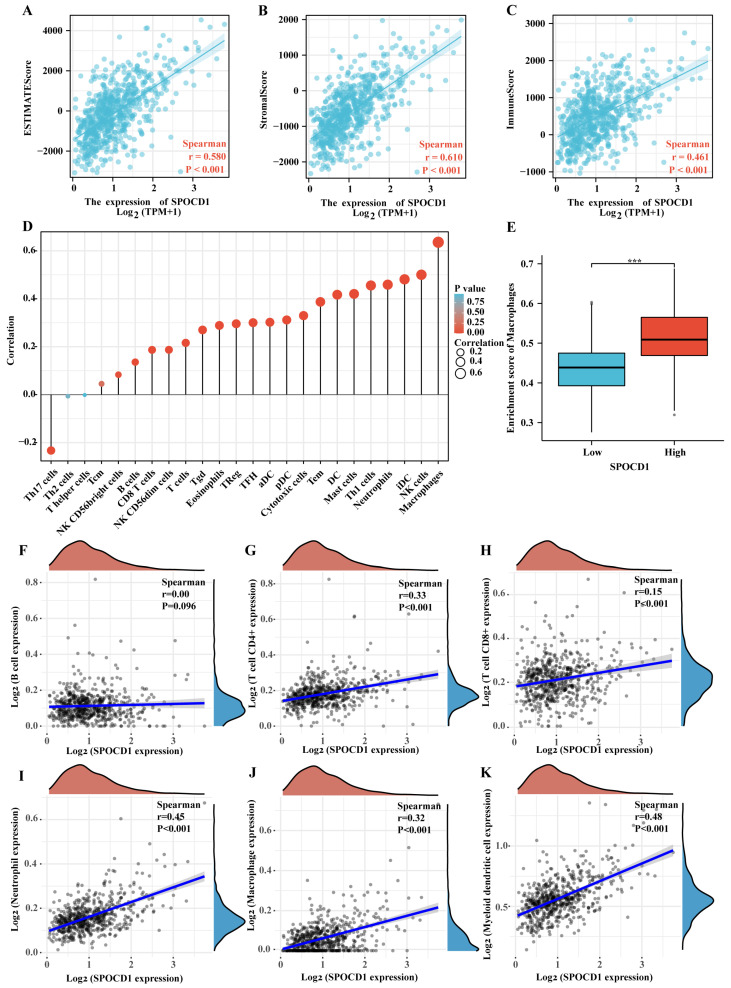
The correlation of SPOCD1 expression with tumor microenvironment and immune infiltration level in CRC. (**A**–**C**) The correlation between SPOCD1 expression and the ESTIMATE, stromal, and immune scores based on the ESTIMATE algorithm. (**D**) The lollipop plot shows the correlation of SPOCD1 expression with immune cell infiltration conducted by ssGSEA. (**E**) Enrichment score of macrophages according to the difference SPOCD1 expressions. (**F**–**K**) The correlation of SPOCD1 expression with immune cell infiltration in CRC acquired from the TIMER online tool. (*** *p* < 0.001).

**Figure 5 biomolecules-13-00209-f005:**
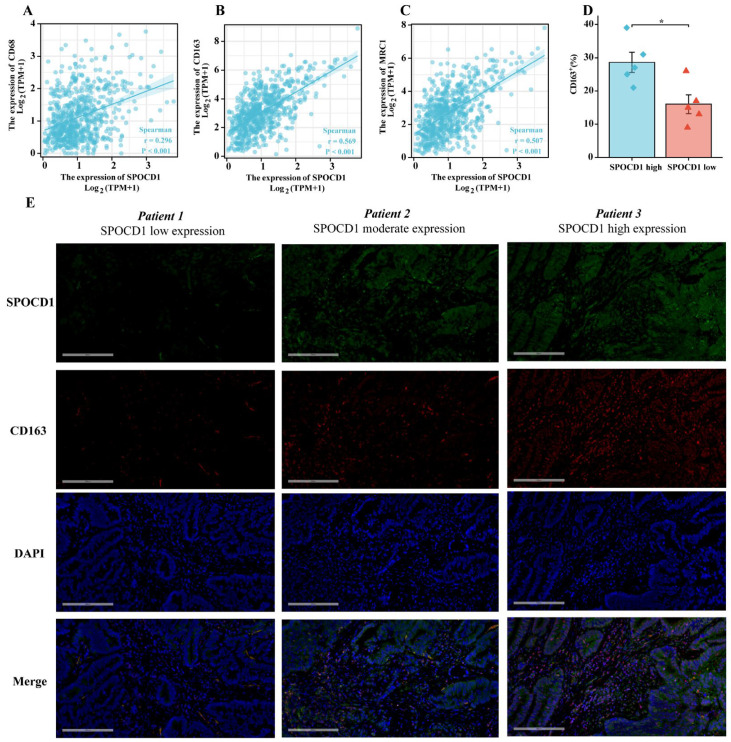
Correlation between SPOCD1 expression level and macrophage markers in CRC. (**A**–**C**) Correlation between SPOCD1 expression and macrophage markers (CD68, CD163, and MRC1). (**D**) CD163 is upregulated in SPOCD1 high-expression group. (**E**) Expression of SPOCD1 and M2 macrophage marker CD163 in CRC tissues (* *p* < 0.05).

**Figure 6 biomolecules-13-00209-f006:**
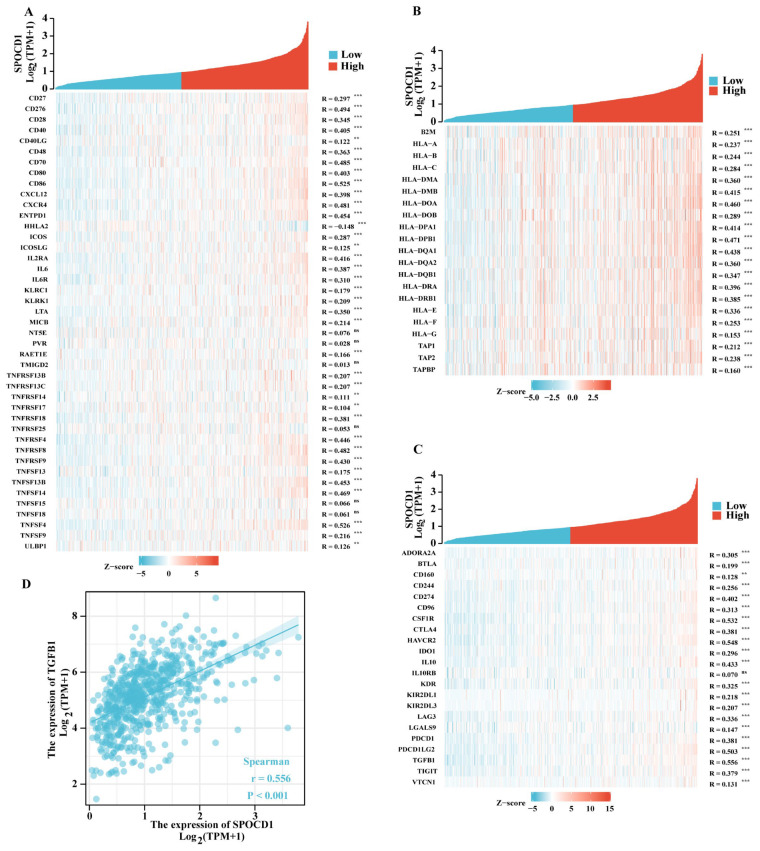
Correlation between SPOCD1 expression level and immunological microenvironment biomarkers in CRC. (**A**) Correlation between SPOCD1 and immunostimulatory gene expression in CRC. (**B**) Correlation between SPOCD1 and MHC molecule expression in CRC. (**C**) Correlation between SPOCD1 and immunoinhibitory gene expression in CRC. (**D**) Correlation between SPOCD1 expression and TGFB1 in CRC (ns represents no significance, ** *p* < 0.01, *** *p* < 0.001).

**Figure 7 biomolecules-13-00209-f007:**
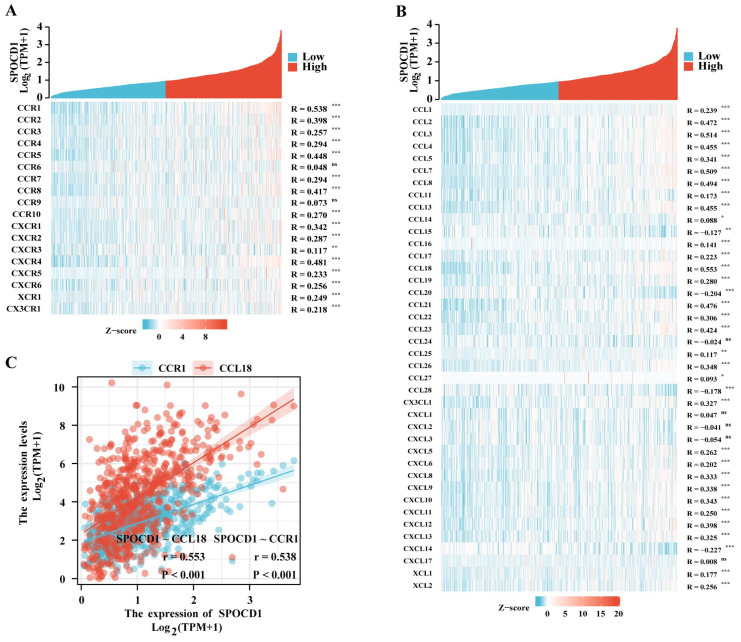
Correlation analysis between SPOCD1 expression and chemokines and chemokine receptors in CRC. (**A**) Heatmap analysis of the correlation between SPOCD1 and chemokine receptors in tumors. (**B**) Heatmap analysis of the correlation between SPOCD1 and chemokines in tumors. (**C**) Correlation between SPOCD1 expression and chemokines CCL18, chemokine receptors CCR1 in CRC (ns represents no significance, * *p* < 0.05, ** *p* < 0.01, *** *p* < 0.001).

**Figure 8 biomolecules-13-00209-f008:**
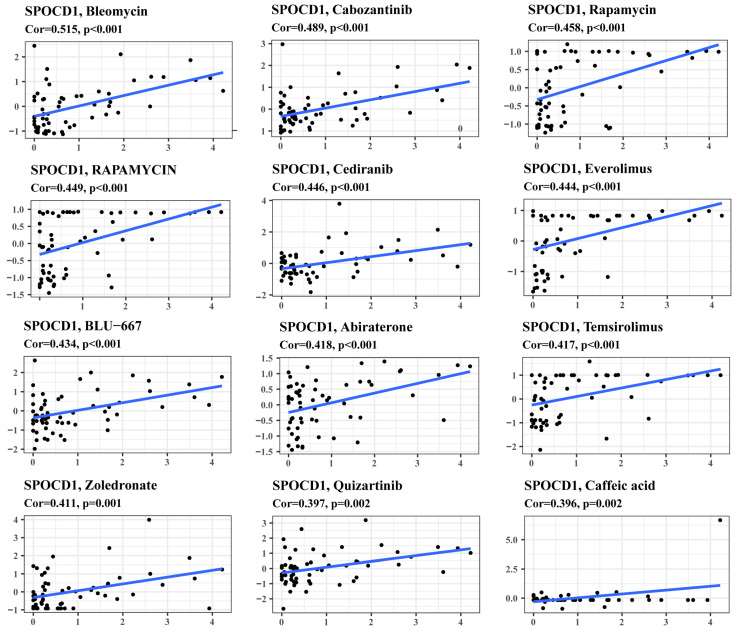
Gene-drug sensitivity analysis based on the CellMiner database: the top 12 drugs with high correlation with SPOCD1 expression in CRC were screened.

**Table 1 biomolecules-13-00209-t001:** The correlation of clinicopathological characteristics and SPOCD1 expression.

Characteristic	Levels	SPOCD1		*p*	Statistic
		Low Expression	High Expression		
*n*		322	322		
Age, *n* (%)	<=65	128 (19.9%)	148 (23%)	0.130	2.29
	>65	194 (30.1%)	174 (27%)		
Gender, *n* (%)	Female	150 (23.3%)	151 (23.4%)	1.000	0
	Male	172 (26.7%)	171 (26.6%)		
T stage, *n* (%)	T1	14 (2.2%)	6 (0.9%)	0.014	10.55
	T2	66 (10.3%)	45 (7%)		
	T3	209 (32.6%)	227 (35.4%)		
	T4	30 (4.7%)	44 (6.9%)		
*n* stage, *n* (%)	N0	193 (30.2%)	175 (27.3%)	0.043	6.29
	N1	79 (12.3%)	74 (11.6%)		
	N2	47 (7.3%)	72 (11.2%)		
M stage, *n* (%)	M0	243 (43.1%)	232 (41.1%)	0.248	1.33
	M1	39 (6.9%)	50 (8.9%)		
Pathologic stage, *n* (%)	Stage I	68 (10.9%)	43 (6.9%)	0.037	8.46
	Stage II	113 (18.1%)	125 (20.1%)		
	Stage III	90 (14.4%)	94 (15.1%)		
	Stage IV	38 (6.1%)	52 (8.3%)		

**Table 2 biomolecules-13-00209-t002:** The univariate and multivariate analysis of Overall Survival (OS).

Characteristics	Total (*n*)	Univariate Analysis		Multivariate Analysis	
		HR (95% CI)	*p*	HR (95% CI)	*p*
Gender	643				
Female	301	Reference			
Male	342	1.054 (0.744–1.491)	0.769		
Age	643				
<=65	276	Reference			
>65	367	1.939 (1.320–2.849)	<0.001	2.858 (1.836–4.450)	<0.001
T stage	640				
T1 & T2	131	Reference			
T3 & T4	509	2.468 (1.327–4.589)	0.004	2.156 (0.981–4.741)	0.056
N stage	639				
N0	367	Reference			
N1 & N2	272	2.627 (1.831–3.769)	<0.001	0.429 (0.165–1.115)	0.082
M stage	563				
M0	474	Reference			
M1	89	3.989 (2.684–5.929)	<0.001	2.278 (1.410–3.682)	<0.001
Pathologic stage	622				
Stage I & II	348	Reference			
Stage III & IV	274	2.988 (2.042–4.372)	<0.001	5.293 (1.816–15.430)	0.002
SPOCD1	643				
Low	322	Reference			
High	321	1.575 (1.109–2.238)	0.011	1.505 (1.022–2.216)	0.038

**Table 3 biomolecules-13-00209-t003:** The univariate and multivariate analysis of Disease Specific Survival (DSS).

Characteristics	Total (*n*)	Univariate Analysis		Multivariate Analysis	
		HR (95% CI)	*p*	HR (95% CI)	*p*
Gender	621				
Female	290	Reference			
Male	331	1.207 (0.769–1.895)	0.412		
Age	621				
<=65	273	Reference			
>65	348	1.421 (0.894–2.257)	0.137		
T stage	618				
T1 & T2	129	Reference			
T3 & T4	489	6.440 (2.029–20.441)	0.002	2.675 (0.812–8.808)	0.106
N stage	617				
N0	358	Reference			
N1 & N2	259	4.119 (2.496–6.797)	<0.001	0.570 (0.219–1.483)	0.250
M stage	542				
M0	455	Reference			
M1	87	7.471 (4.647–12.012)	<0.001	3.834 (2.136–6.881)	<0.001
Pathologic stage	601				
Stage I & II	339	Reference			
Stage III & IV	262	5.716 (3.240–10.083)	<0.001	4.339 (1.324–14.226)	0.015
SPOCD1	621				
Low	307	Reference			
High	314	1.888 (1.190–2.996)	0.007	1.370 (0.843–2.226)	0.203

**Table 4 biomolecules-13-00209-t004:** The univariate and multivariate analysis of Progression Free Interval (PFI).

Characteristics	Total (*n*)	Univariate Analysis		Multivariate Analysis	
		HR (95% CI)	*p*	HR (95% CI)	*p*
Gender	643				
Female	301	Reference			
Male	342	1.217 (0.892–1.660)	0.216		
Age	643				
<=65	276	Reference			
>65	367	1.006 (0.737–1.371)	0.972		
T stage	640				
T1 & T2	131	Reference			
T3 & T4	509	3.198 (1.814–5.636)	<0.001	1.796 (0.991–3.255)	0.053
N stage	639				
N0	367	Reference			
N1 & N2	272	2.624 (1.916–3.592)	<0.001	0.911 (0.389–2.138)	0.831
M stage	563				
M0	474	Reference			
M1	89	5.577 (3.945–7.884)	<0.001	4.375 (2.803–6.827)	<0.001
Pathologic stage	622				
Stage I & II	348	Reference			
Stage III & IV	274	2.924 (2.115–4.044)	<0.001	1.415 (0.547–3.662)	0.474
SPOCD1	643				
Low	322	Reference			
High	321	1.659 (1.215–2.266)	0.001	1.390 (0.998–1.935)	0.051

**Table 5 biomolecules-13-00209-t005:** Correlation between the infiltration level of several types of immune cells and the expression of SPOCD1 in CRC.

Immune Cell Type	Coefficient of Correlation	*p* Value
DC	0.417	<0.001
Activated DC (aDC)	0.302	<0.001
Immature DC (iDC)	0.481	<0.001
Plasmacytoid DC (pDC)	0.312	<0.001
NK cells	0.500	<0.001
NK CD56 bright cells	0.083	0.035
NK CD56 dim cells	0.187	<0.001
B cells	0.136	<0.001
Eosinophils	0.289	<0.001
Macrophages	0.636	<0.001
Mast cells	0.420	<0.001
Neutrophils	0.459	<0.001
Cytotoxic cells	0.330	<0.001
T cells	0.216	<0.001
CD8 T cells	0.187	<0.001
T helper cells	−0.002	0.967
T central memory (Tcm)	0.045	0.251
T effector memory (Tem)	0.387	<0.001
T follicular helper (TFH)	0.300	<0.001
T gamma delta (Tgd)	0.270	<0.001
Th1 cells	0.455	<0.001
Th17 cells	−0.233	<0.001
Th2 cells	−0.007	0.866
TReg	0.296	<0.001

## Data Availability

The data in this study mainly come from public databases, including: Xiantao academic online analysis tool (https://www.xiantao.love/ (accessed on 5 August 2022); The Gene Expression Omnibus (GEO) database: http://www.ncbi.nlm.nih.gov/geo (accessed on 9 August 2022); The Tumor Immune Estimation Resource (TIMER): http://cistrome.shinyapps.io/timer (accessed on 3 September 2022); CellMiner: https://discover.nci.nih.gov/cellminer/home.do (accessed on 15 September 2022). The data of cases from our hospital cannot be disclosed temporarily.

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
