# Peer review of "Prognostic Biomarker SPOCD1 and Its Correlation with Immune Infiltrates in Colorectal Cancer"

_biomolecules, 2023, doi:10.3390/biom13020209_

Round 1

Reviewer 1 Report

1- It would be better the abstract in a better way than the present one.

2- It would be better to rewrite the lines 37-40 in a better way to introduce and define SPOC.

3- The titles of results could be more detailed to represent the result

4- To rewrite the figures legends

5- To rewrite the paragraph 3.4 in a better way to explain the result

Reviewer 2 Report

In this manuscript Gan. et. al have explored the prognostic role of SPOCD1 gene signature in colorectal cancer. There has been several publications that have identified the role of SPOCD1 in colorectal cancer. The authors need to justify their study and discuss these papers.

  1. Fu, Mingyue, Yifan Yuan, Yujuan Mao, Hua Zhu, and Mei Ye. "SPOCD1 accelerates colorectal cancer by increasing the expression of YAP." (2022).
  2. Hui, Juan, Hao Liu, Guangzhou An, Yun Zhou, Junrong Liang, Yangsong He, Pei Wang et al. "SPOCD1 serves as a prognostic marker in colon cancer and is associated with immune infiltration." (2022).
  3. SPOCD1 accelerates colorectal cancer by increasing the expression of YAP https://europepmc.org/article/ppr/ppr574744

  • The authors must elaborate on the parameters utilized for sample size calculation, study power calculation, and effect size calculation.
  • Most of the signatures, both in literature and commercially available like Prosigna or Oncotype DX are multiple gene signatures as single genes have poor accuracy in clinical settings ( most of the studies with single gene are difficult to expand to independent datasets. Authors need to incorporate more than one gene, and term it as a ‘gene signature’ of biologically relevant genes/pathways. The benefit of such an approach is that if a single gene is not showing a strong association, other genes of the same pathway can build on the gaps and deliver a significant gene signature.
  • Multivariate: role of gender and age are absent; the authors must present a complete model.
  • The figure caption for Figure 7 is incomplete.

Reviewer 3 Report

Lin Gan and colleagues investigated the prognostic role of SPOCD1 in Colorectal cancer, also studying the correlations with immune infiltrates in this tumor.

The topic is of great interest, and the study, essentially of bioinformatics, is very accurate and well written.

I have no objections to make, because this study is well organised, well written, the topics covered are discussed with the right consistency and extreme clarity, the bibliography is pertinent and up to date and the conclusions offer a clear overview of the results.

It would have been very interesting to have functional evidence on the biological role of SPOCD, but I hope that, with these comforting results, the authors will be able to deepen their knowledge in their next studies.

Round 2

Reviewer 2 Report

The authors have improved the mansucript.